# Nanocrystalline FeMnO_3_ Powder as Catalyst for Combustion of Volatile Organic Compounds

**DOI:** 10.3390/nano14060521

**Published:** 2024-03-14

**Authors:** Corneliu Doroftei

**Affiliations:** Science Research Department, Institute of Interdisciplinary Research, Research Center in Environmental Sciences for the North-Eastern Romanian Region (CERNESIM), Alexandru Ioan Cuza University of Iasi, Bulevardul Carol I, Nr. 11, 700506 Iasi, Romania; corneliu.doroftei@uaic.ro

**Keywords:** nanocrystalline FeMnO_3_, sol–gel self-combustion, structural properties, catalytic combustion, volatile organic compounds

## Abstract

The paper shows the obtaining of nanocrystalline iron manganite (FeMnO_3_) powders and their investigation in terms of catalytic properties for a series of volatile organic compounds. The catalyst properties were tested in the catalytic combustion of air-diluted vapors of ethanol, methanol, toluene and xylene at moderate temperatures (50–550 °C). Catalytic combustion of the alcohols starts at temperatures between 180 °C and 230 °C. In the case of ethanol vapors, the conversion starts at 230 °C and increases rapidly reaching a value of around 97% at 300 °C. For temperatures higher than 300 °C, the degree of conversion is kept at the same value. In the case of methanol vapors, the conversion starts at a slightly lower temperature (180 °C), and the degree of conversion reaches the value of 97% at a higher temperature (440 °C) than in the case of ethanol, and it also remains constant as the temperature increases. Catalytic combustion of the hydrocarbons starts at lower temperatures (around 50 °C), the degree of conversion is generally lower, and it increases proportionally with the temperature, with the exception of toluene, which shows an intermediate behavior, reaching values of over 97% at 430 °C. The studied iron manganite can be recommended to achieve catalysts that operate at moderate temperatures for the combustion of some alcohols and, especially, ethanol. The performance of this catalyst with regard to ethanol is close to that of a catalyst that uses noble metals in its composition.

## 1. Introduction

In general, catalysts are used when the concentration of combustible gases in the air is very low and cannot support combustion with a flame. Catalytic combustion at low and medium temperatures is used to remove polluting gases from the air, volatile organic compounds (VOCs) such as solvent vapors, mine gases and combustible gases that have escaped from industrial installations, etc. Established and widely used materials such as combustion catalysts are a number of noble metals: platinum, iridium, palladium, rhodium, gold, their alloys with other metals and deposits of such metals on metal or ceramic supports, etc. Recent research on these catalysts aims to improve the methods of their preparation and use [1,2,3,4,5]. The aim is to achieve a large specific area and increase the thermal and chemical stability and the duration of use, for example, by embedding in a ceramic matrix some nanometric particles of metal catalysts [1,2]. The temperature stability of metal catalysts is limited by their volatility [6]. Due to the high cost of noble metals, their use as combustion catalysts is limited especially to industrial processes, where noble metals from spent catalysts are recovered.

In recent years, remarkable catalytic properties have been discovered in a number of oxide compounds: oxides of some metals, ferrites and perovskites, etc.

One of the first oxide catalysts studied was the simple CuCoO_3_ perovskite on a γAl_2_O_3_ support. Compared with a classic platinum catalyst, the elimination of propylene from the exhaust air from the polypropylene processing installations was analyzed. It was found that the new catalyst can successfully replace the platinum catalyst [7].

Another nanostructured perovskite, PrCrO_3_, on a CeO_2_ support, prepared using the urea combustion method, was tested in the post-combustion of diesel engine exhaust gases. The results are similar to those of a platinum catalyst on alumina support [8].

An invention patent proposes their use as combustion catalysts of complex perovskites with rare earths, the recommended compositions being those of the La_x_A_y_B_z_O_3-δ_ type (with oxygen deficiency). The A ion can be strontium, calcium or copper, and the B ion can be cobalt, manganese, chromium or iron. The support, according to the invention, is a complex mineral called mullite (3Al_2_O_3_·2SiO_2_-2Al_2_O_3_·3SiO_2_) prepared in the form of microspheres with a diameter of 20–200 nm and sintered at 1300–1500 °C. The preparation method involves the impregnation of mullite, pressed in the desired form for the catalyst, with solutions of nitrates or acetates of the cations and calcination at 500–700 °C. Thus, effective catalysts were obtained for the depollution of exhaust gases from internal combustion engines, both with gasoline and diesel, at working temperatures below 300 °C [9].

High performances were obtained by a substitution with silver in the perovskite with the composition La_0.88_Ag_0.12_FeO_3_ obtained by grinding in a high-energy mill on an Al_2_O_3_ alumina support. Found were both the reduction of nitrogen oxides and the complete oxidation of hydrocarbons at temperatures of 400–500 °C [10].

Huang et al. prepared La_0.8_Cu_0.2_MnO_3_ and La_0.8_Sr_0.2_MnO_3_ for toluene elimination in the presence of dodecyl-mercaptan, and both catalysts lost activity over time due to the formation of CuSO_4_ or SrSO_4_ [11,12].

Also, a patent proposes the use of manganese–copper spinel ferrite in the form of nanometric particles as a low-temperature combustion catalyst in cigarette filters [13].

A complex oxide compound is described in a patent that proposes a highly active combustion catalyst. A nanostructured cerium-aluminum complex oxide is deposited on an aluminum oxide support, over which a copper-aluminum complex oxide is deposited. The catalyst is able to reduce the nitrogen oxides from the burnt gases [14].

The performance of catalytic combustion is very much influenced by the type of catalyst used. The preparation of catalysts with nanosized particles is the key to an efficient catalytic performance. In nanostructured materials, the interface between nanoparticles and the surrounding medium plays a more important role than in bulk materials [15].

In this work, the porous nanocrystalline iron manganite was synthesized by a sol–gel self-combustion technique using metal nitrates corresponding to the studied compound, and polyvinyl alcohol was used as the colloidal medium [16,17,18]. The catalyst properties were tested in the catalytic combustion of air-diluted vapors of ethanol (C_2_H_5_OH), methanol (CH_3_OH), toluene (C_6_H_5_CH_3_) and xylene (C_8_H_10_) at moderate temperatures (50–550 °C).

## 2. Materials and Methods

Nanocrystalline iron manganite (FeMnO_3_) was obtained using the sol–gel self-combustion method [18] using iron(III) nitrate nonahydrate, manganese(II) nitrate tetrahydrate, ammonium hydroxide solution and poly(vinyl alcohol) (PVA) solution. The solutions of iron and manganese nitrates in stoichiometric quantities were mixed with a solution of poly(vinyl alcohol), forming a colloidal mixture. Ammonium hydroxide solution was added until a soil of metal hydroxides with pH = 8 was formed. The resulting soil was dried at a temperature of 150 °C and then ignited locally, an exothermic combustion reaction taking place. Following the combustion with spontaneous self-propagation, iron manganite resulted in the form of nanocrystalline powder. The powder was subsequently calcined at 500 °C for 30 min, and then it was subjected to a heat treatment in air at 1000 °C for 7 h.

The iron manganite powder obtained after the heat treatment was investigated using thermogravimetric (TG) and differential thermal analysis (DTA) in the temperature range 25–1000 °C at a heating rate of 10 °C/min in static air with X-ray diffraction measurements (XRD), scanning electron microscopy with energy dispersive X-ray spectroscopy (SEM/EDX), X-ray photoelectron spectroscopy (XPS) and measurements using the Brunauer, Emmet and Teller method (BET).

The measurements regarding the catalytic activity for the selected VOCs (ethanol, methanol, toluene and xylene) were carried out at moderate temperatures (50–550 °C) in a flow-type set-up, previously described in ref. [19]. The catalyst in the form of powder (0.5 g) was introduced into a tubular quartz reactor (Ø = 8 mm) with automatic temperature control so that the input gas flow (gas flow rate of 100 cm^3^/min, VOC concentration in air of 1–2‰ and the gas hourly space velocity, GHSV, of 5200 h^−1^) could pass through the entire volume of the powder at a pressure close to atmospheric pressure. The degree of conversion *C* of the gas leaving the reactor was determined as the following [20]:*C* = (*c_in_* − *c_out_*)/*c_in_* × 100 (%)(1)
where *c_in_* is the concentration of the gas at the reactor inlet, and *c_out_* is the gas concentration at the reactor outlet, measured with a detection system with photoionization (PID-TECH) for gases and VOCs. In order to obtain information on the degree of stability of the catalyst, the experiments were carried out both at an increase in temperature and at a decrease in temperature. These determinations were repeated, and similar results were obtained, indicating the stability of the catalyst material over time without its deactivation.

## 3. Results and Discussions

### 3.1. Structure and Morphology

The powder resulting from the self-combustion reaction was investigated using TG-DTA analyses.

From the obtained data (Figure 1), it is evident that the mass of the powder decreases slightly (0.13%) with the increase in temperature up to around 300 °C as a result of the loss of volatile substances.

Corresponding to the temperature of 330 °C, the DTA curve indicates the presence of an endothermic peak that can be interpreted by the production of some crystallization reactions that correspond to a decrease in the powder mass by 0.18%. The crystalline phases of iron manganite begin to appear at temperatures above 900 °C. Thus, one can notice an endothermic peak at 930 °C, corresponding to a considerable decrease in the mass of the powder (2.55%), followed by another endothermic peak at 970 °C where the mass tends to remain constant, indicating the formation of the crystalline structure of iron manganite. Taking into account this information and aiming to obtain a material with good crystallinity, the powder was heat-treated at 1000 °C for 7 h.

The X-ray diffraction measurements made for the heat-treated powder at these parameters confirm the obtaining of iron manganite with a perovskite-type structure with good crystallinity without secondary phases (Figure 2a). The X-ray diffractogram (CuK*_α_* radiation source, wavelength λ = 1.54182 Å) was indexed according to the PDF card No. 75–894 for the iron manganite perovskite. The (211), (222), (123), (400), (411), (332), (422), (431), (125), (440), (611), (026), (145), (622), (134) and (444) peaks reveal that the powder has a cubic perovskite phase with a la3 space group, without any foreign phase. The value obtained for the lattice parameter (a = b = c = 9.40 Å) is close to the values obtained by other authors, (a = 9.40 Å) [21], (a = 9.41 Å) [22,23], who synthesized the iron manganite using different methods. The representation of the spatial structure of the unit cell is given in Figure 2b [24,25,26].

Iron is at site 8b, manganese is at site 24d, and oxygen is placed at site 48e. Manganese and iron occupy 50% of the sites 8b and 24d, respectively; that is, at both 8b and 24d sites, there is equal occupation of manganese and iron (i.e., 50:50) [22,25].

The iron manganite powder obtained by the sol–gel self-combustion route using PVA as a colloidal medium and subsequently heat treated at 1000 °C for 7 h shows a structure with fine granulation (100*–*500 nm) and accentuated porosity in which open pores predominate (Figure 3a*–*c). Also, from the SEM micrographs, a clustered structure of the granules in mini- or macro-agglomerations with irregular shapes and sizes can be highlighted. According to the BET analyses, the specific surface area (S_BET_) and the average crystallite size (D_BET_) have values of 3.20 m^2^/g and 370 nm, respectively (Figure 3d).

Heterogeneous catalysis being a surface phenomenon, its efficiency is determined both by the chemical composition and the structure of the catalyst surface. A nanometric structure ensures a large specific surface area and superior reactivity of the catalyst [27].

The elemental composition and homogeneity of the studied material are confirmed using the energy-dispersive X-ray (EDX) spectra (Figure 3e). In the composition of the material, only the elements Fe, Mn and O are found and can be deduced; the value of the Fe(at%)/[Fe(at%) + Mn(at%)] or Mn(at%)/[Fe(at%) + Mn(at%)] ratio is close to 0.5.

X-ray photoelectron spectroscopy (XPS) using Al-K*α* radiation, is used to investigate the oxidation state of the cations as well as the existence of oxygen vacancies on the surface of the material that play an important role in terms of its catalytic activity [28]. Presented in Figure 4 are the XPS spectra and their deconvolution, attributed to the presence of iron in 2p_1/2_ and 2p_3/2_ (a), manganese in 2p_1/2_ and 2p_3/2_ (b) and oxygen in 1s (c).

The spectrum attributed to iron (Figure 4a) is highlighted by two main peaks located at 724.44 eV (2p_1/2_) and 710.84 eV (2p_3/2_), preceded by satellite peaks located at 732.64 eV and 719.04 eV, respectively. From the deconvolution of the spectrum, it appears that the iron is found in the Fe^3+^ oxidation state, confirmed by the presence of the two satellite peaks that follow the main peaks and the deconvolution peaks located at 727.44 eV and 713.84 eV. Also, iron is also found in the Fe^2+^ oxidation state, confirmed by the deconvolution peaks located at 724.24 eV and 710.64 eV [29,30].

The spectrum attributed to manganese (Figure 4b) is also highlighted by two main peaks located at 652.84 eV (2p_1/2_) and 641.44 eV (2p_3/2_), and from the deconvolution analysis of the spectrum, it appears that manganese is found in the Mn^3+^ oxidation state [31,32].

Also, the spectrum attributed to oxygen (Figure 4c) is highlighted by an asymmetric main peak located at 529.24 eV (1 s); this is specific for transition metal oxides and is also favored by the presence of iron in the Fe^2+^ oxidation state, indicating the presence of oxygen vacancies [33]. The deconvolution analysis of the spectrum reveals the presence of three oxygen species, OH^−^/O^2^, O_2_^2−^/O^−^ si O^2−^, attributed to the peaks located at 532.24 eV, 530.54 eV and 529.24 eV, respectively.

### 3.2. Catalytic Activity

The results of the measurements regarding the catalytic combustion at moderate temperatures of alcohols (ethanol and methanol) and hydrocarbons (toluene and xylene) diluted in air using iron manganite as a catalyst in the form of powder are presented in Figure 5 and Figure 6 and Table 1. Figure 5 shows the degree of conversion as a function of the reaction temperature for catalytic flameless combustion of the studied VOCs.

Typical S-shaped curves were obtained, which describe the variation of the degree of conversion with increasing reaction temperature. These curves indicate that the catalytic activity of iron manganite is much more strongly influenced by the reaction temperature in the case of alcohols than in the case of hydrocarbons. Moreover, the catalytic activity of alcohols begins at higher temperatures than that of the studied hydrocarbons.

The conversion of ethanol vapors starts at 230 °C and increases rapidly around 97% with the temperature increase in the range of 230–300 °C. For temperatures higher than 300 °C, the degree of conversion is kept at the same value. In the case of methanol vapors, the conversion starts at a slightly lower temperature (180 °C) and the degree of conversion reaches the value of 97% at a higher temperature (440 °C) than in the case of ethanol, and it also remains constant as the temperature increases.

For the studied hydrocarbons (toluene and xylene), the conversion starts at temperatures lower than 50 °C. The degree of conversion increases up to the value of 97.5% at 430 °C for toluene and only 60% at 520 °C for xylene, after which it remains constant.

To estimate the catalytic activity of a catalyst, the temperature required for 10% gas conversion (T_10_) when the catalytic reaction is started and is stable is taken into account, as well as the temperature required for 50% gas conversion (T_50_), which is usually chosen as the main indicator of the catalytic activity of a given catalyst [19]. The values of these temperatures are presented in Figure 6 for all tested volatile organic compounds. The lowest T_10_ temperatures are obtained in the case of hydrocarbons, 110 °C and 190 °C for toluene and xylene, respectively, followed by alcohols, where the values of these temperatures are very close, 230 °C and 240 °C for methanol and ethanol, respectively.

At T_50_, the catalytic activity is high enough, and the interactions between the catalyst surface and the reactants are intense. It can be stated that the iron manganite catalyst obtained using the method mentioned above has a high catalytic activity for ethanol, methanol and toluene, while for xylenes, it is much lower. The performance of this catalyst is remarkable with regard to ethanol (Figure 5); it approaches that of a catalyst that uses noble metals in its composition. Deng Qian et al. [34] carried out a study on the catalytic combustion of ethanol using catalysts based on Au/*γ*-Fe_2_O_3_. With such a catalyst, a conversion value of 99.6% can be achieved for ethanol at a temperature of 290 °C. Also, a study carried out by Avgouropoulos et al. [35] regarding the catalytic combustion of ethanol using catalysts based on Pt/Al_2_O_3_ highlights a conversion value of 90% at a temperature of 220 °C.

The catalytic activity of the FeMnO_3_ catalyst in the conversion of ethanol, methanol, toluene and xylene can be largely attributed to the mobility of oxygen inside the perovskite lattice due to the oxygen vacancies generated by the presence of iron ions with variable valence, Fe^2+^/Fe^3+^, as it could be ascertained from the analysis of the XPS spectrum (Figure 4). Oxygen vacancies favor the appearance of oxygen ion species adsorbed on the iron manganite surface. Although the mechanism regarding the catalytic activity of oxide compounds with a perovskite structure is still under debate, according to some widely accepted opinions, at low temperatures (<400 °C), the catalytic activity of oxide compounds with a perovskite structure in total oxidation reactions of gases is, to a large extent, determined by the amount of weakly bound surface oxygen species [15,36], which in the present case are assumed to be much more available for the oxidation of alcohols (ethanol and methanol) compared to those for the oxidation of hydrocarbons (toluene and xylene). The weaker the oxygen bond on the catalyst surface, the more active the catalyst is for the complete oxidation of gases [15,36,37]. Of course, the roles of specific surface area and particle size are very important. A large specific surface area that can be obtained by an appropriate preparation method of nanomaterials, in the present case the sol–gel self-combustion method, will involve much more interactions between the iron manganite surface and the tested VOCs and, therefore, a high catalytic activity towards their combustion.

The apparent activation energies *E_a_* of the catalyst in the combustion reactions of ethanol, methanol, toluene and xylene were determined using Arrhenius curves (the natural logarithm of the reaction rate constant k at low conversion, below 15%, as a function of the inverse of the absolute temperature, 1/T) [15]. This graph is a straight line, and from its slope, the apparent activation energy was calculated [15,37].

Table 1 shows the values of the degree of conversion at two temperatures, 290 °C and 500 °C, as well as the values of the kinetic parameters (apparent activation energy and reaction rate) for the oxidation of VOCs studied by the FeMnO_3_ catalyst. The values of the apparent activation energies are higher in the case of alcohols and lower in the case of hydrocarbons (Table 1) [38,39,40,41,42,43,44,45]; also, the catalytic activity starts at much higher temperatures in the case of alcohols compared to hydrocarbons (Figure 5).

The reaction rate normalized to the specific surface area can characterize the specific catalytic activity. A catalyst is more active the higher the reaction rate. The reaction rate changes substantially from 1.07 × 10^−2^ μmol s^−1^m^−2^ to 14.25 × 10^−2^ μmol s^−1^m^−2^, depending on the gas type. The highest value of the reaction rate (14.25 × 10^−2^ μmol s^−1^m^−2^) was obtained for the combustion of ethanol by the studied catalyst; it demonstrated the best catalytic activity in the combustion of ethanol (a 95.43% degree of conversion at 290 °C, reaching 97.5% for temperatures higher than 300 °C). The lowest value of the reaction rate (1.07 × 10^−2^·μmol s^−1^m^−2^) was obtained for the combustion of xylene, the catalyst also showing the weakest catalytic activity in xylene conversion (a 59% degree of conversion at 500 °C), as can be seen in Figure 5.

The values obtained for the kinetic parameters in the case of the studied oxidic compound as a catalyst for ethanol, methanol, toluene and xylene vapors are comparable to the values reported in the literature for a series of oxidic compounds with a similar structure and VOCs [39,40,41,46].

From the obtained results, it can be stated that the porous oxide compound with a perovskite-type structure (FeMnO_3_) obtained using a sol–gel self-combustion method can be a promising candidate for the realization of catalysts that operate at moderate temperatures for the combustion of some alcohols, especially ethanol.

## 4. Conclusions

The paper shows the obtaining of iron manganite (FeMnO_3_) in the form of a nanometric powder with high purity and crystallinity and its investigation from the point of view of its catalytic properties for a series of volatile organic compounds (ethanol, methanol, toluene and xylene) at moderate temperatures (50–550 °C).

The conversion of ethanol vapors starts at 230 °C and increases rapidly around 97% with the temperature increase in the range of 230–300 °C. For temperatures higher than 300 °C, the degree of conversion is kept at the same value. In the case of methanol vapors, the conversion starts at a slightly lower temperature (180 °C) and the degree of conversion reaches the value of 97% at a higher temperature (440 °C) than in the case of ethanol, and it also remains constant as the temperature increases.

In the case of the studied hydrocarbons (toluene and xylene), the conversion starts at temperatures lower than 50 °C. The degree of conversion increases up to the value of 97.5% at 430 °C for toluene and only 60% at 520 °C for xylene, after which it remains constant.

The studied iron manganite can be recommended to achieve catalysts that operate at moderate temperatures for the combustion of some alcohols, especially ethanol. The performance of this catalyst in terms of ethanol approaches that of a catalyst that uses noble metals in its composition.

## Figures and Tables

**Figure 1 nanomaterials-14-00521-f001:**
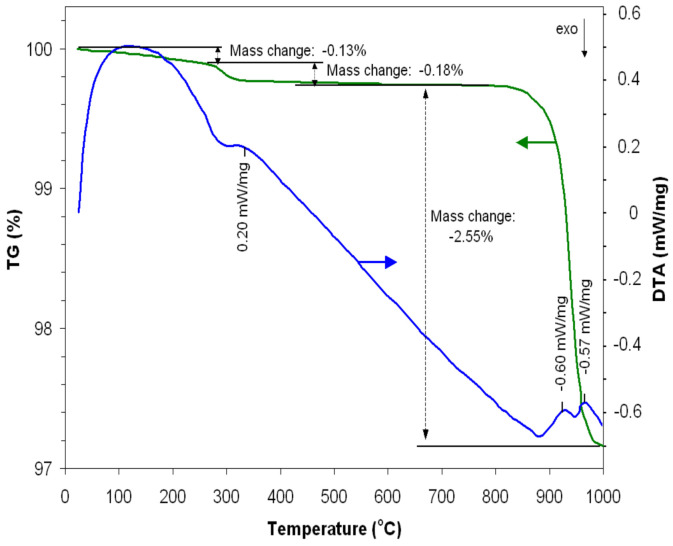
TG-DTA of synthesizing iron manganite.

**Figure 2 nanomaterials-14-00521-f002:**
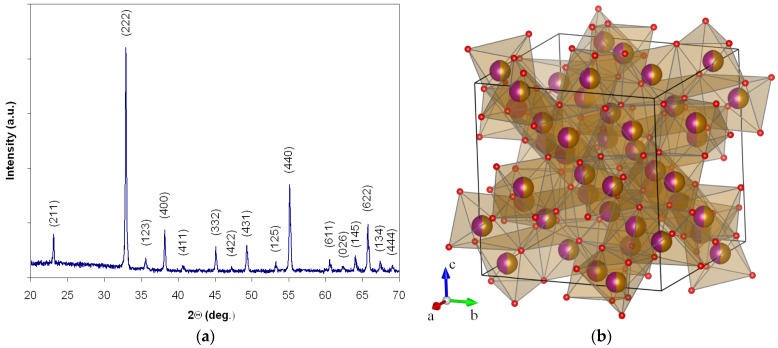
The X-ray diffractogram (**a**) and 3D representation of the unit cell in *la3* symmetry for iron manganite (**b**). Iron ions are orange, manganese ions are purple halves of the balls, and oxygen ions are red balls.

**Figure 3 nanomaterials-14-00521-f003:**
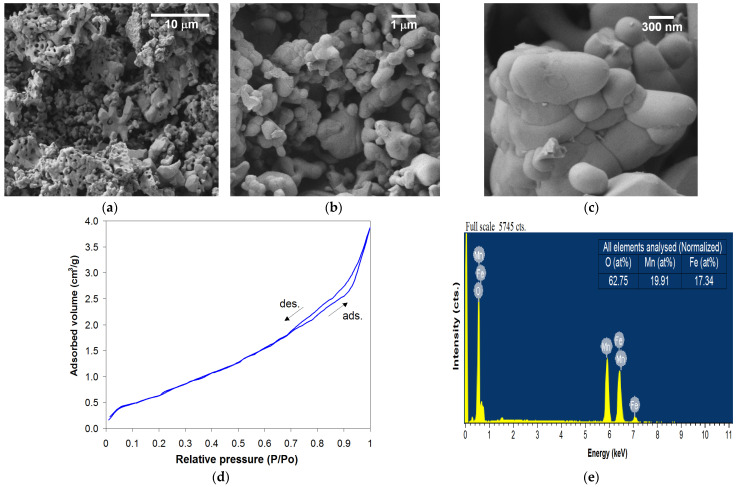
SEM micrographs obtained at different magnifications (**a**–**c**), BET adsorption–desorption isotherm (**d**) and EDX spectrum with the analyzed elements (**e**) for iron manganite heat-treated at 1000 °C/7 h.

**Figure 4 nanomaterials-14-00521-f004:**
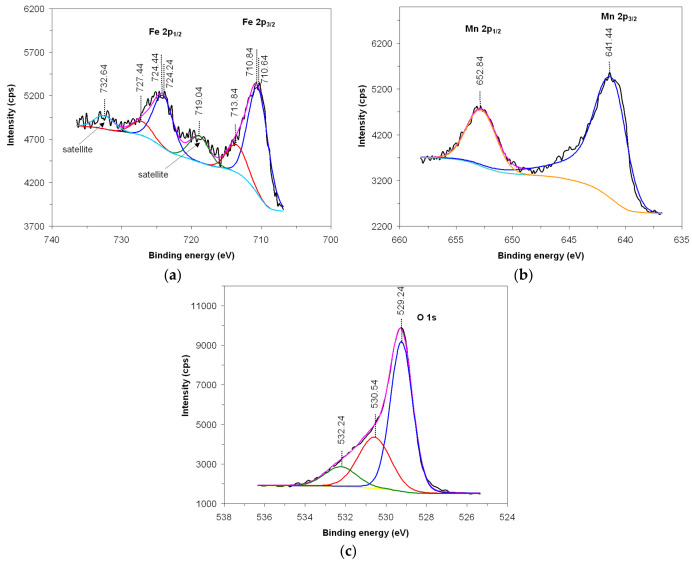
XPS spectrum of Fe 2p (**a**), Mn 2p (**b**) and O 1s (**c**).

**Figure 5 nanomaterials-14-00521-f005:**
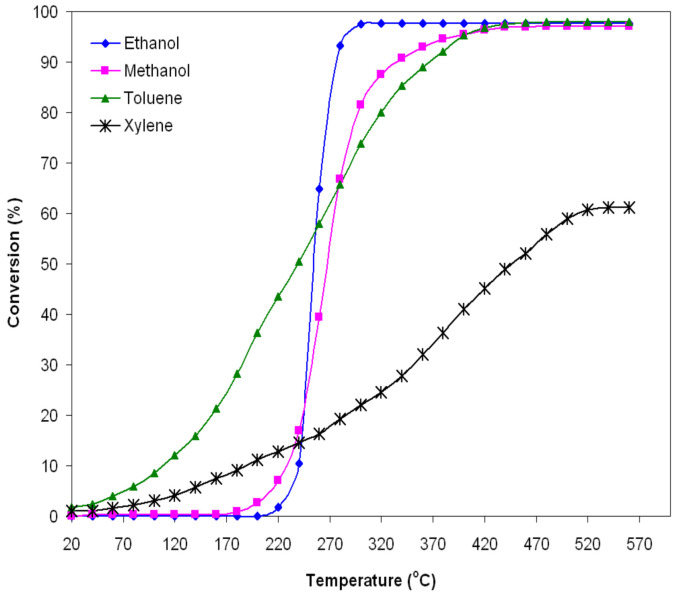
Conversion of ethanol, methanol, toluene and xylene as a function of the operating temperature of the FeMnO_3_ catalyst (1–2‰ VOC in air, 0.5 g catalyst, and GHSV = 5200 h^−1^).

**Figure 6 nanomaterials-14-00521-f006:**
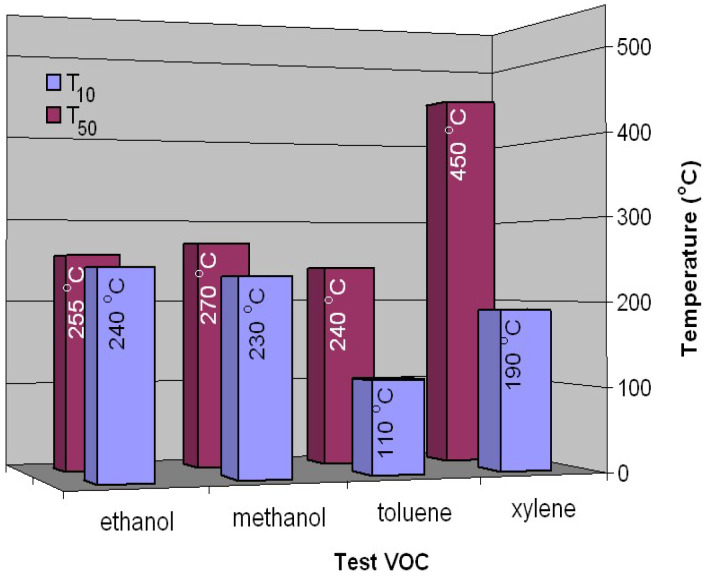
T_10_ and T_50_ temperatures versus tested VOCs for the FeMnO_3_ catalyst.

**Table 1 nanomaterials-14-00521-t001:** Parameters of the degree of conversion and kinetic parameters.

VOCs/. FeMnO_3_	Conversion at 290 °C (%)	Conversion at 500 °C (%)	Reaction Rate * (µmol s^−1^ m^−2^)	Activation Energy ** (KJ/mol)
Ethanol	95.43	97.58	14.25 × 10^−2^	195.248
Methanol	74.09	97.05	6.27 × 10^−2^	99.816
Toluene	70.25	97.91	5.55 × 10^−2^	19.477
Xylene	20.63	59.00	1.07 × 10^−2^	18.232

* Reaction rate (at 290 °C) for VOC concentration at low conversion per unit surface area of catalyst. ** Apparent activation energy for low conversions.

## Data Availability

The data presented in this study are available on request from the corresponding author.

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
