# Peer review of "Nanocrystalline FeMnO3 Powder as Catalyst for Combustion of Volatile Organic Compounds"

_nanomaterials, 2024, doi:10.3390/nano14060521_

Round 1

Reviewer 1 Report

Comments and Suggestions for Authors

The manuscript is in general in a good shape with proper experimental design and reasonable results. Some minor comments are given in there.

a) In line 12 of introduction, the authors mentioned "starts at slightly higher temperature". It is suggested that to clarify the temperature range referring to slightly higher temperature.

b) Add some detail of TGA experiment, such as the gas used, the heating rate. etc. 

c) For activity measurement, what is debit 100 cm3/min? The concentration 1-2 ‰. Why the concentration is not constant?

d) The authors are suggested to add bit more discussion regarding the  discussion of mechanism. For example, Line 323-325, it was suggested "due to the amount of reactive oxygen species on the surface of the catalyst"? how do you think the role of oxygen transfer from bulk to catalyst surface? what is the role of Fe and Mn in the oxygen transfer?

Comments on the Quality of English Language

The English is easy to be understood. 

Reviewer 2 Report

Comments and Suggestions for Authors

In this paper, the acquisition of high-purity and highly crystalline iron manganese oxide (FeMnO3) and the investigation of its catalytic properties were conducted with a focus on performance at moderate temperatures (50–550 ℃) for volatile organic compounds (ethanol, methanol, toluene, xylene). The studied iron manganese oxide is recommended to achieve catalysts operating at moderate temperatures for the combustion of some alcohols, especially ethanol. The catalytic performance of this catalyst for ethanol is comparable to catalysts using noble metals in their composition. The insights gained from this study are deemed significant in the field of catalysis. Therefore, I recommend the acceptance of this manuscript. Authors are encouraged to address the following concerns:

  1. It might be beneficial to mention not only catalysts of metals and alloys but also metal oxide catalysts, such as PdO, IrO2, PtO2, from line 33 to 35.
  2. A more detailed explanation or illustration of the polyhedron in Figure 2b would be helpful. Understanding the structure in Figure 2b is challenging.
  3. The mention of "VESTA [24,25]" seems unrelated to VESTA. Please clarify this reference.
  4. In Table 1, Xylene shows the lowest reaction rate, but why does it have the lowest activation energy for the reaction? If the activation energy is low, the reaction rate will be higher.
